# Mapping Processing Elements of Custom Virtual CGRAs onto Reconfigurable Partitions

Zbigniew Mudza * and Rafał Kiełbik

Department of Microelectronics and Computer Science, Lodz University of Technology, ul. Wólczańska 221, 93-005 Łódź, Poland; rkielbik@dmcs.pl
* Correspondence: zmudza@dmcs.pl

**Abstract:** FPGAs can provide application-specific acceleration for computationally demanding tasks. However, they are rarely considered general-purpose platforms due to low productivity of software development and long reconfiguration time. These problems can be mitigated by implementing a coarser overlay atop the FPGA fabric. Combining this approach with partial reconfiguration allows for the modification of individual processing elements (PEs) of the virtual architecture without altering the rest of the system. Module relocation can be used to share implementation details between functionally equivalent PEs that use identical sets of resources, thus eliminating redundant placement and routing runs. Proper floorplanning is crucial for virtual Coarse-Grained Reconfigurable Architectures (CGRAs) with relocatable PEs considering their tendency to use nearest-neighbor connection patterns. It requires solving two problems—finding identical regions in the FPGA fabric and assigning individual partitions to certain locations. This article presents minor improvements of a state-of-the-art solution for the first and proposes a novel technique for solving the other. The proposed automated floorplanner uses modified breadth-first search with direction-based penalties to create initial floorplan consistent with geometry of logical array, then improves the result with 2-opt local optimization. Compared to simulated annealing solutions, the proposed approach allows for the reduction in the floorplanning time by two to three orders of magnitude without compromising the quality of the results.

**Keywords:** Reconfigurable Computing; FPGA; partial reconfiguration; virtual architectures; CGRA; design methodology; floorplanning; productivity



## 1. Introduction

The breakdown of Dennard scaling [1] made satisfying the ever-growing demand on computational power increasingly difficult. The limited range of further performance improvements (power wall, multicore bottleneck) and relatively low energy efficiency of General-Purpose Processors (GPP) has caused a great interest in alternative computing approaches.

### 1.1. Coarse-Grained and Fine-Grained Reconfigurable Logic

The idea of the Reconfigurable Computing (RC) [2] paradigm is to adjust hardware architecture to fit application requirements. Instead of decomposing a function into a series of basic operations performed on fixed hardware, custom temporary data-paths optimized specifically for this purpose are generated. In theory, such an approach combines performance and energy efficiency of hardware with flexibility of software.

Field Gate Programmable Arrays (FPGAs)—the most popular reconfigurable platforms, use fine-grained Look-Up Table (LUT) based reconfigurable logic that allows for the definition of independent custom formulas and connections for individual bits. As a result, FPGAs can be used to emulate virtually any digital circuit. However, this incredible flexibility comes at a cost. Each reconfigurable component introduces area and energy consumption overhead compared to non-reconfigurable counterparts. The complexity of

hardware with multiple configuration options for all primitives is drastically increased, which requires reducing clock frequency to achieve timing closure. Moreover, considering that not only functionality but also placement and routing of each individual gate-level operation is determined independently, both logical and physical synthesis of FPGA designs are difficult and time-consuming tasks. Furthermore, having more configuration options leads to larger configuration data size and consequently longer reprogramming times. The introduction of partial reconfiguration [3], which allows modifying functionality of some components while preserving the configuration of the rest, helped to mitigate this problem. There is however another issue: since many applications consist mostly of arithmetic operations, the flexibility of fine-grained logic often cannot be fully exploited. In order to achieve better efficiency for byte- or word-based computations, modern FPGAs combines fine-grained logic with interwoven coarser blocks such as DSPs and RAMs. For arithmetic calculation intensive applications, reconfigurable platforms based solely on such blocks can be even a better fit.

CGRAs (Coarse-Grained Reconfigurable Architectures) is a loosely defined group of programmable devices with processing elements (PEs) of coarser spatial granularity operating on larger arguments (e.g., 32-bit words) rather than individual bits. The architectures proposed over the years vary in purpose, degree of reconfigurability, granularity level, construction of functional units, and interconnection types. As a result of the transition to a smaller number of more complex processing elements, common functionality changes for entire operands, and bus-based connection routing, CGRAs sacrifice flexibility but offer different advantages: superior performance for arithmetic operations, reduced energy consumption, and faster reconfiguration. What is more, the higher abstraction level and simplified architecture can be much more convenient for reconfigurable systems end users, especially if they lack hardware background.

### 1.2. Virtual Coarse-Grained Reconfigurable Architectures

Due to the lack of mature, commonly available physical CGRA architectures and both enormous costs and extremely long development of custom ASIC CGRA, virtual architectures implemented as overlays atop FPGAs are not uncommon [4–15]. Apart from serving prototyping purposes for architecture development, Virtual Coarse-Grained Reconfigurable Architectures (VCGRAs) can be used to provide a convenient high abstraction layer for FPGA-based systems end users. As the underlying hardware is still fine-grained, the approach does not offer performance gains, energy efficiency improvements, nor reductions of space overhead in silicon. Nevertheless, it can be used to simplify the system description and narrow down the design space, therefore reducing the development cycle (develop, implement, debug) time, which is crucial whenever time-to-market is essential.

Mapping an application onto an FPGA with an intermediate overlay layer comprises two distinctive tasks: front-end (application onto virtual architecture) and back-end (overlay onto physical FPGA) mapping. Considering that the overlay architecture is only temporarily formed within the FPGA fabric, the entire virtual architecture can be altered and remodeled if it benefits mapping or execution. In other words, there are two levels of reconfigurability—adjustments of VCGRA configuration for minor and entire overlay replacement for major changes. Functional changes of the coarse blocks can be performed by using register-based circuit switching within a single FPGA configuration, dynamic partial reconfiguration for block content swapping, or a combination of the two. The latter approach offers a virtually unlimited number of possible PE variants and resource utilization overhead reduction, but changing functionality takes much longer as it requires reprogramming a section of FPGA. It also entails binding overlay blocks to certain regions of the underlying chip.

Xilinx FPGAs use module-based partial reconfiguration (also referred to as Dynamic Function Exchange). Top-level logic is divided into a set of individually reprogrammable modules and a static part. On a higher abstraction level the reconfigurable partitions can be treated as processing elements of a CGRA. Normally, each instance of a functional

module uses individual, location-specific implementation. Consequently, an array of N × M identical PEs requires N × M implementations for each configuration. However, it is possible to force relocatability of the modules, so that common implementation results could be used regardless of location.

This study presents a methodology for mapping custom coarse-grained overlay architectures with fully reconfigurable processing elements onto Xilinx FPGAs. The innovative aspect of the proposed approach is that it uses reconfigurable partition based virtual coarse blocks with module relocation used to reduce the number of location-specific implementations. The proposed design flow can be divided into the following stages:

- Definition of virtual coarse block interface;
- Specification of its resource requirements;
- Identification of potential partition regions in the target FPGA chip;
- Floorplanning of virtual coarse blocks;
- Implementation of the static part of the architecture with black-box modules;
- Implementation of any desired functionality in a relocatable reference coarse block.

The methodology allows coarse-grained blocks array scaling and easy migration between different physical platforms. Although the proposed methods were developed for Xilinx 7-Series devices, they should be easily adaptable to Xilinx UltraScale FPGAs, as the architectural differences are mostly low level.

## 2. Related Work

### 2.1. Trends in Coarse-Grained Reconfigurable Architectures Design

The authors of [16] present a very thorough review of more than thirty different Coarse-Grained Reconfigurable Architectures, outline general trends, and propose a classification. They define CGRAs as architectures characterized by spatial reconfiguration granularity at fixed functional unit level or above and temporal reconfiguration granularity at region/loop-nest level or above. Despite significant differences between individual architectures, some common characteristics can be observed. Typically, CGRAs use a 2D mesh or direct-connection based network of PEs with 32-bits wide operations, static (compile-time) scheduling, and to some extent dynamic (run-time) reconfiguration of both computing elements and interconnections. These features are used as a CGRA design reference for the purpose of architecture analysis within the scope of this study. Virtual architectures in the form of FPGA overlays are not explicitly mentioned in the review. However, not only do they meet the proposed CGRA definition, but some of the analyzed architectures (e.g., [17,18]) use FPGA overlay implementation as a prototype.

A more recent survey [19] covers recently emerged architectures. The authors suggest that, despite significant progress, CGRA architectures and tools have not yet reached sufficient maturity for a widespread commercial use. The work analyzes technical trends, outlines challenges, and presents a novel multidimensional taxonomy for CGRAs. The proposed architecture specification uses abstract high-level programming, computation, and execution models to describe the CGRA from a behavioral perspective. At a lower level, micro-architecture defines technical details of actual physical manifestation of processing elements, interconnects, etc. The authors argue that virtualization—i.e., operating on a unified micro-architecture independent model to generate common abstract configurations that can later be interpreted onto specialized physical CGRAs—is the most crucial step towards widespread adoption of the Coarse-Grained architectures. It is suggested that state-of-the-art FPGA virtualization techniques could provide good foundations for physical CGRA-targeted virtualization. In particular, FPGA-based CGRA overlays are raised as a reference point. They are also considered to be a feasible solution in systems using physical FPGAs, as the abstract programming layer offers platform and CAD independence, agile development, and general productivity increase.

### 2.2. Virtual Coarse-Grained Reconfigurable Architectures

For all the benefits they offer, CGRAs have one fundamental flaw—they are not generally available. As a result, virtual implementations in the form of FPGA overlays are often used to exploit some of the advantages of CGRAs. In addition to its undoubted value in prototyping and architecture development, the overlay approach has an advantage from the programming perspective, even if it does not provide the same performance nor energy efficiency [20].

Multiple architectures, analysis, techniques, and tools for Virtual CGRAs have been introduced over the years [4–15,21]. In spite of the lack of standardization, different motivations and approaches to virtual CGRA architecture design, common features, and solutions can be observed.

Typically, a CGRA overlay uses a 2-D mesh of homogeneous processing elements (PEs) with NESW (north-east-south-west) connections [4–7,21]. A linear communication pattern is often used for image processing applications [8]. More complex models like crossbar [9] and multistage global interconnections [10] offer simplified front-end mapping, but high connection cost and potential congestion problems make them unusable for large networks.

The micro-architecture of individual PEs is rarely discussed. In most cases, the VCGRA design operates on higher abstraction and implementation details and depends entirely on FPGA CAD tools responsible for back-end mapping. The few exceptions include arrays of DSP-based processor-like cores [11,22]. Reconfigurability of the virtual architecture layer is usually obtained by using a runtime circuit switching controlled by setting registers and virtual switch blocks implemented using FPGA logic resources. However, in order to reduce resource utilization overhead, tunable LUTs and switch blocks were proposed [13,23,24] to utilize FPGA configuration memory for that purpose.

In contrast to CGRAs with physical manifestation, overlay architectures can offer another layer of reconfiguration as the entire virtual architecture can be reformed. QUKU [4], a prototype FPGA-based overlay with 2-D mesh of virtual programmable elements, is a good example. On the FPGA configuration layer, each of the PEs can be configured as one of several possible variants, like a multiplier or a simple ALU. On the overlay layer, a simple instruction-like configuration is used to determine a cycle-by-cycle behavior of individual PEs (which input arguments are used, which parts of the available operation are performed in ALU, etc.). The proposed solution offers rapid overlay configuration changes, which are essential for temporal distribution of computations. However, this requires reprogramming the entire FPGA if even a single PE is replaced with a different module.

It is worth noting that although the most overlays use homogeneous coarse PE arrays, they are often used as a component of heterogeneous systems. Most commonly, VCGRAs are used with general-purpose CPUs, not only as external hosts controllers but often in the form of soft cores implemented in FPGA [7] or processors embedded with FPGA in a SoC [6,22]. Moreover, as the granularity is defined by the abstract programming model, fine-grained logic can be used alongside it to provide better adaptability and state machine logic handling in a so-called Mixed-Grained or Multi-Grained approach [15,20,25].

A recent survey [26] comparing multiple VCGRA architectures suggests that overlays can be effectively used as general-purpose, on-demand application accelerators for FPGA-based systems. They do introduce a significant hardware performance penalty, but in the case of general-purpose acceleration, adaptation to application changes is more important. According to the study, this is where overlays really excel, with three orders of magnitude reduction in placement and routing time as well as hardware kernel context switching time.

### 2.3. Mapping Tools and Techniques

Apart from work focused on specific VCGRA architectures, many design tools and techniques that can be used with a variety of coarse-grained overlays have been proposed recently. The majority of the work uses standard off-the-shelf FPGA flows for the back-end and concentrates on different aspects of front-end mapping. Focusing on application mapping is understandable since temporal granularity of application-specific configura-

tions is much finer than that of overlays. Consequently, as multiple applications can be mapped and run using a single virtual architecture, the impact front-end mapping has on performance and especially development productivity is significant.

CGRA-ME [21,27] is an extremely versatile, VTR [28] inspired mapping framework for CGRA that can be used with both physical and overlay manifestations. It allows for the definition of custom architectures via an XML-based description language and supports a variety of connection patterns. The framework translates LLVM-compiled intermediate representation of a program into a data-flow graph (DFG) and maps it onto PEs of the architecture using a simulated-annealing based mapper or integer linear programming technique. Virtual architecture can then be implemented as either standard-cell or FPGA overlay using standard tool flows. The presented virtual solution uses Altera/Intel FPGAs and tools. It is mentioned that placing a physical realization of PEs in the form of a regular grid coherent with logic representation in the virtual CGRA is desired, but no methods for obtaining that goal are presented.

A huge advantage of overlays is that each application can be mapped onto an individual optimized version of virtual architecture. QuickDough [6], a design framework for soft CGRA accelerators, analyses requirements of compute kernel and selects the best fitting accelerator from a library of pre-built overlays. The library can be updated upon users' requests with custom architectures based on the provided SCGRA template. The framework uses standard FPGA design flow to implement the overlays, but the authors note that due to regularity of SCGRA template architectures, other state-of-the-art techniques [12] (described further in this section) could be used to accelerate the process.

Prioritizing front-end mapping without considering its influence on the back-end process can have a negative impact on VCGRA performance. Even though front-end mapping is theoretically independent from the low-level aspects of the micro-architecture, individual techniques may be applicable only for architectures with certain structures and interconnection networks or they may work well only for applications with specific communication patterns. Certain architecture features that simplify the front-end mapping can increase the complexity of the back-end process. Crossbar routing with direct connections between all PEs is an exemplary case. It is an excellent solution for Just-in-Time task scheduling, as it simplifies the overlay placement and routing problem to O(1), at the expense of the connection cost of $O(N^2)$ [9]. Apart from direct changes in required routing resources, increasing complexity diminishes the chances of CAD tools finding a solution with satisfactory timing and no congestion.

Increasing the efficiency of the overlay to FPGA mapping and improving the performance of the actual underlying hardware are less common subjects of research, as standard techniques and tools for logical synthesis and implementation for FPGAs are considered efficient. However, specificity of VCGRA architectures can be exploited for better results.

TLUT/TCON [13,23,24] is a complex design flow for FPGA overlays aimed at reducing logic utilization overheads. It uses the parametrized VCGRA model with expression-based definitions of setting registers which are then implemented as tunable look-up tables (with table entries being Boolean functions of a parameter rather than ones and zeros). When an application is mapped onto such an overlay, appropriate parameters' values are determined and applied to the placed and routed generic configuration in a so-called specialization stage. This way, setting registers are kept entirely in the configuration memory of the FPGA instead of using additional logic resources. A similar approach is proposed for changing settings of individual routing switches; however, this is possible only for hypothetical FPGAs, as vendors do not provide access to low level routing reconfiguration infrastructure. It is worth noting that parametrized configuration generation and specialization require an entire dedicated toolchain developed for this purpose.

Despite being identical to the overlay perspective, individual PEs of virtual architectures are usually implemented independently for better global optimization. According to LaForest and Steffan [5], partitioning can be used as a simple solution to improve compute density and speed of mesh based overlays, as it prevents unnecessary deduplication of

components in the critical paths. Moreover, the implementation result of a single instance can be reused at other locations. Rapid Overlay Builder (ROB) [12] proposes a combination of techniques for fast implementations of virtual architectures in Xilinx FPGA, without compromising the efficiency of the solution. Instead of performing independent placement and routing for each PE, ROB uses hard macro relocation of implementation results, with possible multiple variants for different sets of resources. The flow comprises resource budgeting for PEs, floorplanning their distribution in FPGA fabric, placing and routing initial PE variants, relocating implementation results, and interconnecting the design. However, the two initial phases require manual engagement from the users. The proposed techniques can be used to obtain significant speedup of placement and routing process, but as they are based on now obsolete ISE and GoAhead tools, the flow is incompatible with newer Xilinx FPGAs.

Recently introduced IMPRESS [14,15] is an academic design flow for Xilinx FPGAs capable of mapping virtual architectures onto networks of reconfigurable partitions. The tool is based on set of TCL scripts working under Xilinx Vivado. It exploits module relocation to reduce implementation time and bitstream memory footprint. Although proper floorplanning of the reconfigurable partition is claimed to be crucial for efficient mapping of virtual architectures, no automation for the process is provided.

### 2.4. Relocatable Partitions in FPGAs

Modern FPGAs are highly heterogeneous and modules can only be relocated between partitions using compatible sets of resources. Reusing the placement and routing result at another location by applying them as fixed design constraints can speed up implementation and hence shorten the development cycle [29,30]. Nevertheless, relocating entire partial bitstreams is far more advantageous. Even though many studies on this subject were published over the years, a vast majority (i.e., [31–36]) have become outdated with the emergence of new FPGA architectures and changes in design tools.

Oomen et al. [37] established a great foundation for bitstream relocation in Xilinx 7-Series devices. Their work presents design requirements and methods that can be used to enforce relative placement and routing of modules in order to achieve bitstream compatibility. Unfortunately, the work oversees potential global routing feed-through and can lead to breaking routing connections if it occurs. Retkowski et al. [38] proposed a very similar approach extended by a feed-through prevention obtained by module isolation. Both studies are presented as proof-of-concept and address solely the relocation between two compatible partitions.

For the purpose of relocating virtual coarse-grained block implementation results (as constraints or bitstreams) to multiple locations within the same or across multiple designs—as presented in this study—the authors proposed an improved technique based on the aforementioned work. Its distinctive features are:

- Separated workflows for structural static design and reconfigurable module functionalities;
- Nested sub-PBlock based constraining of partition-to-static port buffers;
- Cross region routing prevention (as adjacent regions can be unavailable at different positions);
- Local routing loop-back prevention (as loop-back resources replacing connections in the available direction occur only in boundary located switchboxes).

This study focuses on different aspects of Virtual CGRA implementation in FPGAs; partition relocatability is only presented to the extent required to comprehend the proposed solutions. Technical details of all methods and tools used to obtain reconfigurable module relocatability are described in full in a previous publication by the main author [29,39].

Another similar relocation technique has been recently proposed as a part of the IMPRESS design automation tool [14,15]. The main difference between the approaches is in the global routing feed-through prevention method. Instead of applying the vendor provided module isolation mechanisms, IMPRESS generates blocker nets around reconfigurable regions.

### 2.5. Partition Floorplanning

The majority of state-of-the-art FPGA partition floorplanning solutions (i.e., [40–44]) focus on optimizing area coverage (best utilization of available area, with minimal overhead) and minimizing total wire length (Manhattan distance between connected components). Typically, optimization is based on simulated annealing. However, some solutions use Mixed-Integer Linear Programming, which is claimed to be more effective for large partitions with a time limited search. Contrary to the previously used techniques, all contemporary solutions must consider FPGAs heterogeneity. Another key restriction for floorplanning reconfigurable modules in Xilinx devices, acknowledged in the aforementioned work, is regularity of partition geometry (due to the configuration memory construction, only rectangular reconfigurable regions aligned to clock region borders are supported).

Module relocation entails even more restrictions as it is only possible if original and target regions use identical resource distribution. Given the heterogeneity of modern FPGAs, partition relocatability is only obtainable at very specific locations in the fabric. Consequently, the characteristics of the floorplanning problem are completely different than for designs without module relocation. Instead of being an area coverage problem, floorplanning becomes a combination of common pattern search (finding regions with identical resource distribution) and combinatorial optimization (selecting which region should be occupied by each PE).

R. Backasch et al. [45] proposed a technique that can be used to identify homogenous reconfigurable regions in modern heterogeneous FPGAs for the purpose of partition relocation. The approach exploits column-based regularity of resource distribution in Xilinx FPGAs. The technique identifies column patterns that satisfy resource requirements for partition, eliminates non-minimal patterns, and finds all occurrences of regions that use such a distribution in a target chip. The authors present search results for Xilinx Virtex (xc5vfx70, xc6vlx240, xc7vx690) devices but do not test the partition relocation between such regions. Although the general ideas are relevant and constitute a solid foundation for the work presented in this study, several issues important for actual implementation of relocatable partitions were not addressed. Regions might be misidentified as compatible since no differentiation between resource columns of opposite alignment is made. Inseparability of adjacent interconnection tiles is also not respected, which may cause back-to-back violations in Xilinx 7-Series [46]. Finally, no isolation feasibility checks are performed.

AutoReloc [36] is a bitstream relocation design-flow enhancing capabilities of the now obsolete Xilinx PlanAhead software. It uses a two-phase partition floorplanning algorithm. The initial stage is a column pattern choice. Contrary to [45], it uses a different search strategy and considers left- and right-alignment of resource tiles and interconnection columns inseparable. The proposed search method always finds only minimal patterns, which is usually an advantage, but, as explained later in this article, is not always desirable. What is more, the algorithm omits non-reconfigurable regions, but can place partitions directly adjacent to them, which can make partition-to-static routing generated elsewhere impossible to recreate, unless additional restrictions are applied. The technique could be adapted to use with module isolation, required for routing feed-through prevention, but does not support it in the original form. From all patterns found with at least a designer-specified number of occurrences, one is selected based on arbitrary but reasonable criteria. Once a pattern is chosen, simulated annealing is performed to find optimal selection of available blocks. The objective function is defined so that total distance between selected regions is minimized and minimal distance between two partitions is bigger that an empirically predefined threshold (to avoid congestion). The optimization is limited to selecting the best set of reconfigurable regions; no effort to select the optimal position for individual partitions is made.

Contrary to the presented state-of-the art solutions, floorplanning techniques proposed in this study support module isolation and optimize placement of individual partitions.

## 3. Materials and Methods

Exploiting reconfigurability of the underlying FPGAs enhances capabilities of Virtual CGRAs in both prototyping and end-use applications. FPGAs can be reprogrammed to change configuration of setting registers [23], swap processing elements with different variants [4], or even replace entire overlay [6]. However, reconfiguring the entire FPGA is time consuming and might be inefficient if changes are limited. Partial reconfiguration (PR) allows for the reprogramming of a section of FPGA, while leaving the rest of the fabric untouched. Due to vendor-specific construction of configuration memory, partial reprogramming affects fixed sectors of FPGA primitives, grouped (differently across various devices) based on their physical location in the chip. Therefore, virtual components of an overlay architecture need to correspond to specific sets of resources in the underlying chip, to be individually reconfigurable on the FPGA level.

### 3.1. Resource Distribution and Reconfigurable Partition in Xilinx FPGAs

Xilinx FPGAs support module-based partial reconfiguration under a name of Dynamic Function Exchange (DFX). At the top level of hierarchy, any Partially Reconfigurable design is divided into a number of reconfigurable partitions and a static part. Each partition needs to be assigned to a partition block (PBlock)—a PR compatible sector of the FPGA fabric. At the top level, all partitions are treated as black-boxes; any reconfigurable module with compatible interface can be implemented within the partition, provided that PBlock resources are satisfactory for module requirements.

Modern Xilinx FPGA architectures use highly heterogeneous resources. For the purpose of PBlock construction analysis, only reconfigurable logic resources are of relevance, other resources (including communication transceivers, input–output buffers, PCIe controllers, XADC converters, etc.) will be omitted. Xilinx FPGAs use coarse Block RAMs and DSPs alongside two variants of fine-grained LUT-based Configurable Logic Blocks—with (CLBM) and without (CLBL) additional distributed RAM and shift registers. The internal structure of each Xilinx FPGA device is highly connected to the construction of its clock tree. The fabric is divided into clock regions corresponding to horizontal clock routing channels branching from a centrally located vertical clock spine or multiple vertical spines in the case of UltraScale devices. Resources are grouped by type and distributed unevenly in columns perpendicular to clock region rows (Figure 1). In general, resources of a single type fill an entire span of a column within a single clock region—occasional occurrences of non-reconfigurable resources and empty spaces in the FPGA fabric prevent some columns from being uniform across multiple clock rows. In both Xilinx 7-Series and UltraScales, routing resources used by the components are placed alongside either side of the logic alternately, in separate interconnection block columns (INT). Despite being uneven, the distribution is very regular with observable repetitiveness of identical patterns (Figure 1).

The smallest unit of configuration memory that can be independently reprogrammed is defined as a configuration frame. It contains configuration data for all components in a clock-region wide section of a single logic column and the routing corresponding switchboxes. Configurations of individual components reside at fixed-offset positions, based on their relative vertical location within the span of the clock region. A partition block can cover area corresponding to multiple configuration frames tiled horizontally (within a clock region) and vertically (across multiple regions), but only rectangular shapes are allowed. Furthermore, enlarging the partition area increases time required to reprogram it. Theoretically, reconfiguration of a single frame is allowed, but as partial reconfiguration flow in Xilinx Vivado tools does not allow separating adjacent interconnection tiles (back-to-back violation) [46], it is only feasible at specific locations.

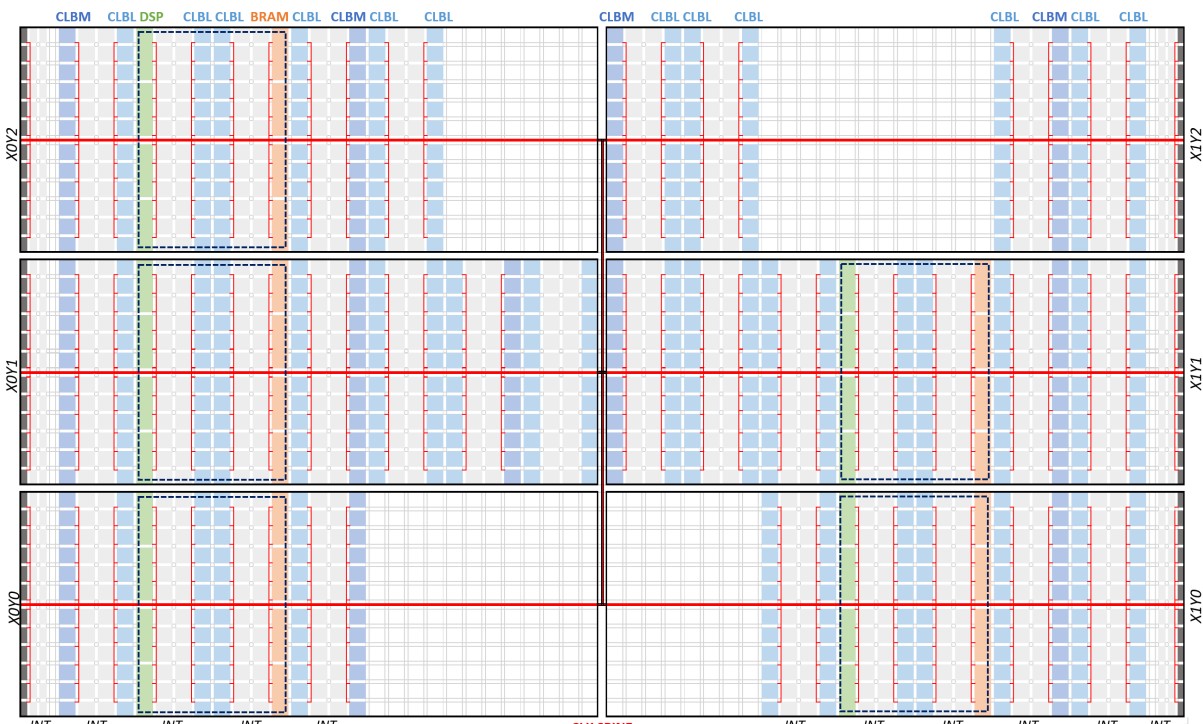

**Figure 1.** Simplified schematic of resource distribution in Xilinx 7-Series devices. The FPGA fabric is divided into lock regions (X0Y0 to X1Y2), interconnection blocks are located to either side of logic resources (CLB, BRAM, DSP). Dashed lines indicate repetitions of arbitrarily selected column pattern across clock regions.

### 3.2. Overlay Architecture Assumptions

In theory, any CGRA can be implemented in the form of an overlay atop sufficiently powerful FPGA, but due to the characteristics of the underlying hardware, virtual implementations of some architectures might be far less efficient than others. In addition, the diversity of CGRAs causes great differences in both front-end and back-end mapping tasks, hence establishing universal techniques compatible with all architectures is impossible. The mapping methodology presented in this study is relatively versatile, it allows using various architectures based on the most popular solutions, and it proposes reconfigurable partition usage that can maximize its impact on the VCGRA operation. Nevertheless, it is based on certain assumptions and restrictions regarding overlay architecture (presented further in this section).

At a high abstraction level, CGRA systems can be described as arrays of coarse-grained processing elements connected to host processors, memory, and peripheries (Figure 2). Depending on the architecture, CPU might be used only for configuration or during system operation as control unit or main processing platform (with CGRA being an auxiliary accelerator). In the case of VCGRA, CPUs embedded in FPGAs—in the form of either soft cores (e.g., Microblaze, NIOS) or SoC hardware (e.g., ARM core in Zynq)—can be used instead of external processors. Additional virtual hardware, including reconfigurable fine-grained logic, might also be implemented in the FPGA alongside the PE array. For the purpose of the proposed mapping methodology, only the PE array is considered. Other components may either be distributed across the FPGA as a part of global static design or assigned to certain locations. In the latter scenario, occupied regions need to be defined before partition floorplanning to eliminate them from the allowed search space.

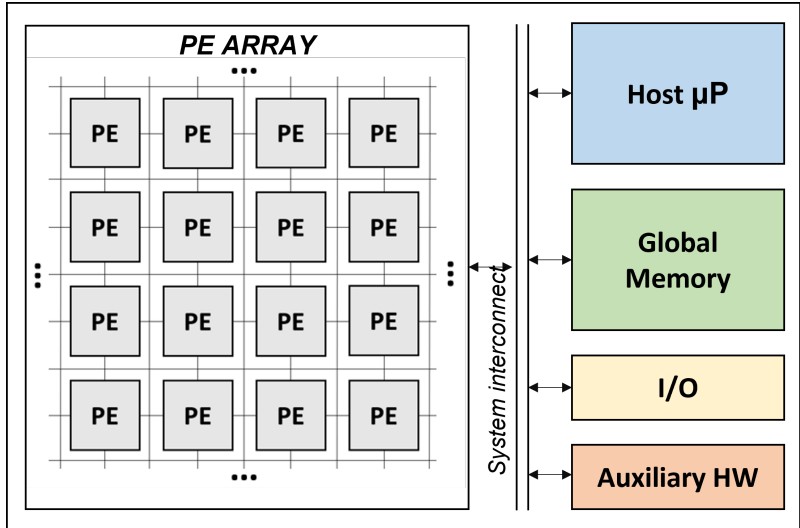

**Figure 2.** General model of CGRA systems. Host processor might be used only at configuration stage (standalone CGRAs) or throughout execution stage (for system control as well as computations).

Due to relatively large size of minimal independently reconfigurable fragment of FPGA fabric, partial reconfiguration is suitable for significant functional changes in processing elements rather than connection switching. At the same time, using a small number of large partitions limits reconfiguration time reduction, depriving the approach of its greatest advantage. In order to fully exploit partial reconfiguration, partitions should be used as core components for a large number of PEs. It needs to be stressed that the abstract overlay structure does not need to match the actual physical partitioning (Figure 3). Static components of processing elements can be excluded from reconfigurable partitions in order to reduce their size, hence decreasing reprogramming time. Moreover, virtual switches of interconnection network corresponding to individual PEs, but regarded as separate entities at the overlay level, can be included within the scope of reconfigurable partitions to reduce logic utilization overhead. Due to restrictions of the PR mechanism, the reconfigurable modules must be instantiated at the top level of design hierarchy. As a result, such solutions require non-intuitive architecture description. Therefore, direct PE-to-partition binding might be more convenient.

Even though proposed model supports multitude of interconnected network structures and allows setting-register-based run-time switching, some solutions are better suited for use with the proposed approach than others. Mesh-based patterns can be effectively used with a large number of elements. Direct connections between a small number of neighboring partitions simplifies routing in the FPGA, which can reduce net delays and probability of congestion problems. Moreover, 2-D mesh is the most popular connection pattern in both virtual and physical CGRAs [16,19]. Typically, only connections to the nearest PEs in four basic NESW (North-East-South-West) directions are used but expanded patterns with additional connections are not uncommon (Figure 4). The proposed methodology is designed to work with 2-D mesh based VCGRAs. The current version of the objective function used in the floorplanning procedure assumes only basic NESW connections. If the architecture uses additional connections, calculations of their cost need to be appended to the objective function.

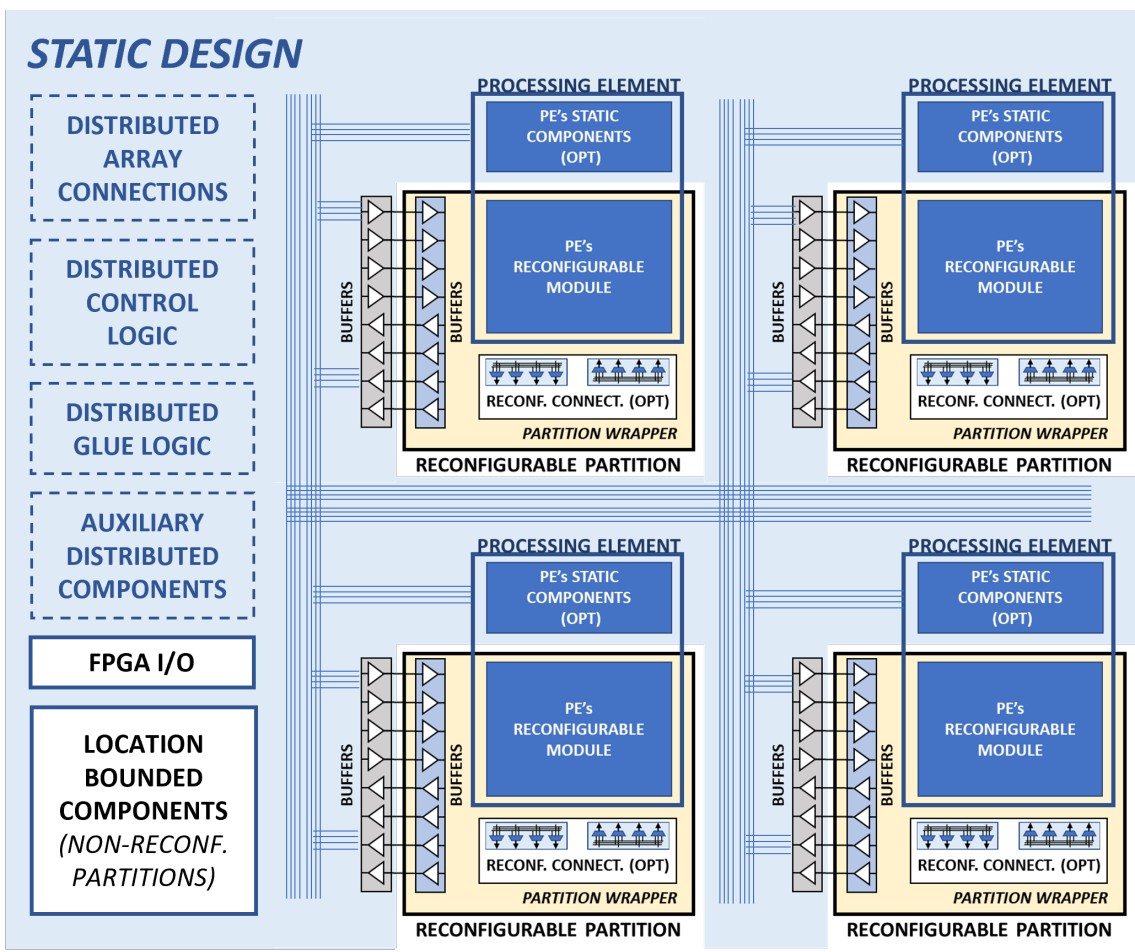

**Figure 3.** Proposed model of Virtual CGRA using partial reconfiguration mechanism for PE functionality changes.

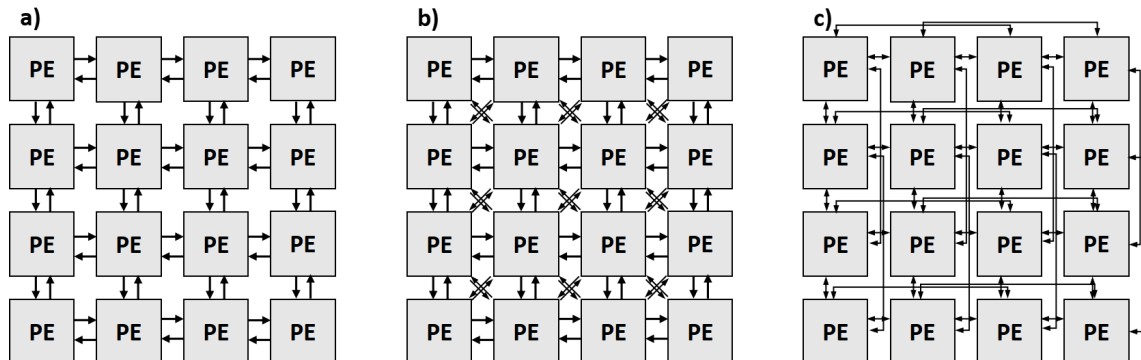

**Figure 4.** Mesh-based communication patterns with direct NESW (North-East-South-West) connections between PEs (**a**) and expanded versions with additional diagonal connections (**b**) and extended distance NESW (**c**).

The proposed floorplanning procedure uses techniques based on exploiting 2-D mesh network characteristic to accelerate the process. Significant modification might be required to produce satisfactory results for other popular connection patterns (Figure 5). In VCGRAs with direct connections between any PEs (e.g., crossbar connections), interconnection cost grows with the square of PE number. Considering that the proposed approach assumes using many processing elements, it is not well suited to such connection patterns.

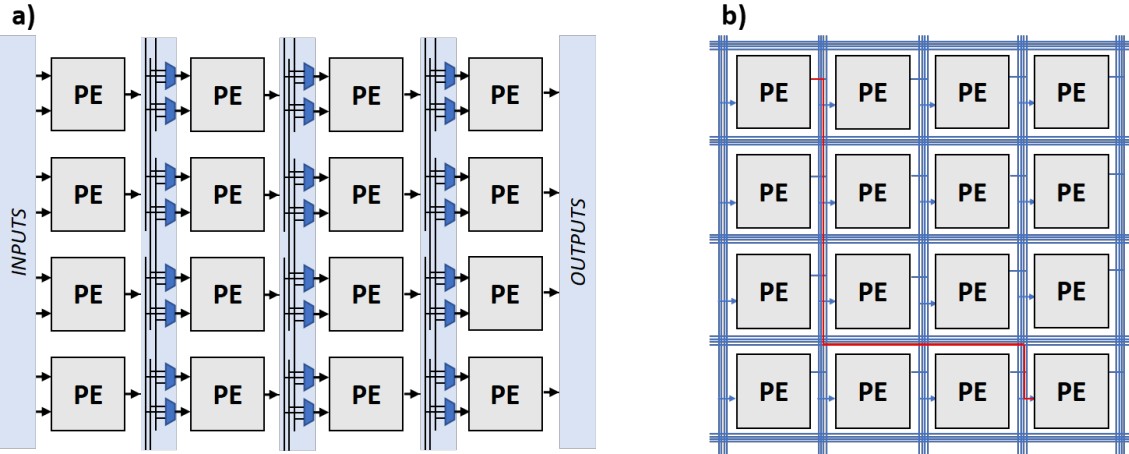

**Figure 5.** Other patterns used in Virtual CGRAs: linear dataflow (**a**) and configurable each-to-any (e.g., crossbar) connection (**b**).

### 3.3. Module Relocation

In the proposed approach, homogenous array of processing elements uses numerous instances of logically equivalent partitions. Physical implementation of any module associated with a partition needs to be contained within explicitly defined sector of FPGA fabric—PBlocks. If partitions are assigned to compatible PBlocks with the same arrangement of hardware components, they can use identical relative placement and routing (P&R). Instead of performing independent implementation runs for each instance, common results can be applied to all partitions.

Module relocation—i.e., moving a fully placed and routed module to another partition at a different position—can be performed by forcing fixed P&R constraints or by readdressing entire partial bitstreams. In the first scenario, location-specific implementation runs are drastically accelerated since modules are already pre-placed and pre-routed. In the other, the location-specific run is not required at all, provided that timing closure can be guaranteed by different means (e.g., by using register buffers or constraining net delays on both sides of partition-to-static connection) [29].

Although Xilinx Vivado toolchain offers no support for module relocation per se, it can be obtained using a combination of inbuilt Partial Reconfiguration Flow (PRF) and Isolation Design Flow (IDF) with additional constraining and TCL scripted processing of intermediate results. The techniques used to provide module relocatability are described in detail in previous work by the main author of this article [29,39]. General concept and details with crucial influence on floorplanning are outlined below.

Partition relocation is performed between partitions that have vertical span of a single clock region and identical set of reconfigurable resource columns with the same interconnected alignment. For each different PBlock construction, reconfigurable modules are implemented only once in a so-called reference partition. Placement and routing results are then extracted and re-applied as constraints with new location offsets. Alternatively, a partial bitstream can be generated and relocated by changing frame address register value. Static design with black-box partition is implemented independently. Partitions are isolated from the static to prevent global routing feed through. A so-called isolation fence—single primitive wide space that cannot be used for logic placement—is required. Partitions can be assigned to the same columns in adjacent clock regions. However, as global routing cannot cross partitions, at least a clock-region high sector of all columns needs to be included in the static. Partition-to-static connection cannot cross clock-region borders, as the adjacent regions may be unavailable at different locations. Consequently, partition ports must be placed only alongside either left, right, or both vertical borders. All partition-to-static nets use additional buffers on both ends to preserve identical connection and help with timing closure management. Buffers in the static serve as anchoring points. They can be

implemented alongside different entities in CLBL or CLBM columns. The column type does not need to match, but the placement in relation to partitions must be the same. This is obtained by implementing the partition-to-static interface in the reference design and then relocate the results in form of P&R constraints. The entire process is fully automated and does not require any manual engagement.

### 3.4. Design Flow

Imposing the aforementioned assumptions and restrictions made it possible to develop a productivity-oriented back-end mapping methodology for Virtual CGRAs with relocatable reconfigurable partitions, compatible with off-the-shelf FPGA hardware and CAD tools. The presented solution is an updated and refined version of the methodology proposed in the earlier work [29], adapted for the specificity of virtual CGRA designs and expanded with new functionality—most notably automated floorplanner. In this approach a VCGRA system description with PE array of parametrizable size and explicitly defined reconfigurable partition instances is divided into static and reconfigurable parts that use separate design flows (Figure 6).

Static design with user-defined MxN PE array and black-box partitions is synthesized using standard Xilinx Vivado tools. The proposed partition floorplanner searches through target FPGA for, preferably homogeneous, regions that satisfy resource requirements of the PE partitions. Then, it attempts to find optimal locations for individual partitions in a reasonably short time. The obtained floorplan is then used to implement the design using Xilinx Vivado toolset and module relocation methodology [29,39]. The same parametric static overlay can be implemented with different floorplans using various PBlock constructions and PE array sizes or targeting different FPGA. A library of implemented reconfigurable overlays can store different designs for future use.

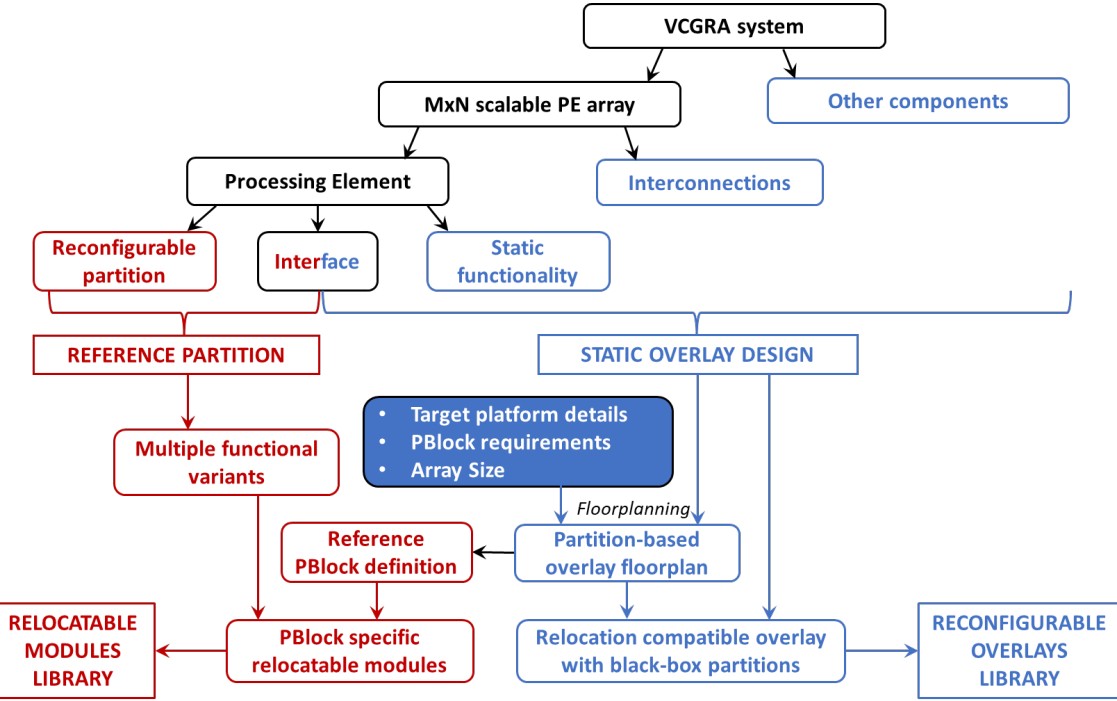

**Figure 6.** VCGRA with relocatable partitions—back-end mapping design flow overview.

Multiple functional variants of modules are developed and implemented separately, in a so-called reference partition design, using the module relocation methodology. As the relocatable modules are location independent, a single variant of placed and routed module for a given PBlock construction can be used for all instances in multiple floorplans and FPGA platforms. If a floorplan uses several types of PBlocks, one reference design of each

type is required. On the basis of the application characteristics and target platform capacity, the most appropriate application-specific combination of static and reconfigurable modules can be selected during front-end application mapping (Figure 7).

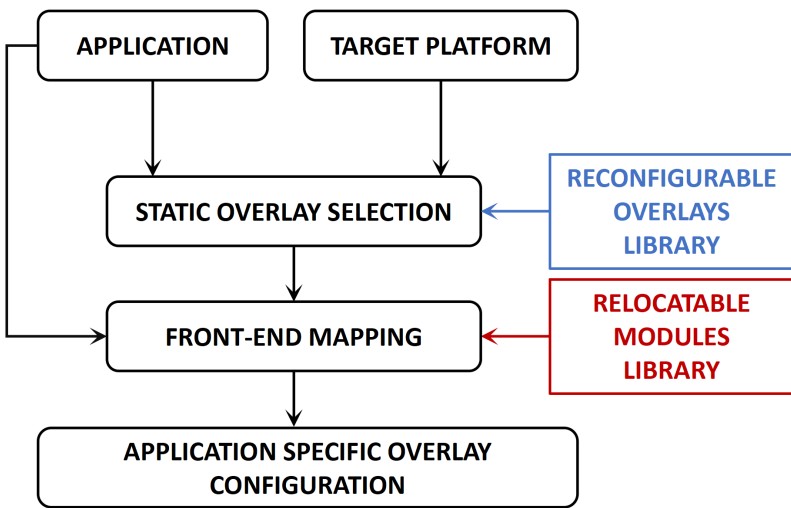

**Figure 7.** VCGRA with relocatable partitions—front-end mapping design flow overview.

The main contribution of this study is an automated custom floorplanner for Virtual CGRAs with relocatable reconfiguration partitions. It is a TCL-based tool that expands the design flow proposed in earlier work [29], which formerly required manual floorplanning. The toolchain was developed for Xilinx FPGAs and operates as an extension for vendor provided software (executed from Vivado command prompt). However, the original floorplanning approach presented in this article could potentially be adapted to different hardware and software.

The importance of proper floorplanning is significant, considering the fixed binding between the reconfigurable partitions and regions of the underlying FPGA. In order to maximally reduce the number of PE implementation variants used in the design, the floorplanner should attempt to assign PEs to as many identical PBlocks as possible, as module relocation can only be performed between compatible regions. Moreover, directly connected processing elements should be located in close proximity to each other in order to reduce delays and eliminate routing congestion problems.

Finding optimal floorplan is an NP-hard problem that requires completing two tasks:

1. Finding appropriate partition block candidates—regions in FPGA fabric with enough resources to fit required logic with minimum overhead—for each partition;
2. Selecting partition locations (PE to PBlock candidate assignments) capable of producing best routing results (short net delays, no routing congestion), optimal design density, and the lowest number of different implementations of PEs.

### 3.5. Finding Relocation Compatible Regions

The first step in finding appropriate regions for reconfigurable partitions is determining resource requirements. If all possible variations of reconfigurable modules are known in advance, the number of primitives of each type should be determined as a superposition of variants' resource utilization, estimated during logical synthesis, with an additional margin. However, with the proposed separation of VCGRA structure and module functionality development, modules' functionality might be unknown at that stage. In that case, several generic overlay implementations with PBlocks using different configuration of resources can be prepared beforehand and the most appropriate variant can be selected once actual module requirements are determined. Considering that relocatable partitions operate on entire columns within the scope of a single clock region, resource requirements based on

primitives are converted to a minimal number of columns of different types that need to be included within PBlocks.

The state-of-the-art solutions [36,45] can identify homogeneous regions with specified amount of reconfigurable resources. However, additional conditions need to be satisfied for regions to support module relocations:

- Regions with compatible column types but opposite alignment must be differentiated;
- Partition borders cannot be placed between inseparable adjacent interconnection columns;
- Feasibility of isolation fence insertion and anchoring point placement needs to be checked.

The proposed search procedure addresses these issues. The approach obtains resource distribution information directly from Xilinx Vivado software (using get_tiles and get_property queries with filters) and performs column pattern checks in individual clock regions. Multiple non-overlapping regions of different constructions are considered because the number of identical regions can be smaller than the required PE array size. The procedure comprises the following steps:

1. Define partition candidate search parameters:

    - Minimal number of columns with given resource type (DPSs, BRAMs, LUTs etc.)—user provided or calculated from primitive-based resource utilization estimation. Note that certain resources are contained within more than one column type;
    - Maximum column width of partitions;
    - Anchoring points position (left/right/either/both sides of PBlock);
    - Overlapping pattern reduction preferences (see steps 10–12);
    - Optional prohibited regions, excluded from PBlock search space.

2. Select target platform.
3. Obtain clock regions information for the target. Start search for a single region.
4. Identify indices of columns where resources of certain types occur and store them in temporary lists.
5. Reduce search space based on availability of BRAMs and DSPs.

    - BRAMs and DSPs instances in FPGA fabric are quite rare. If several columns of BRAM or DSP types are required, the search can be limited to few locations where such columns occur within acceptable range.

6. Identify PBlock candidates—subsectors of search space with required resources.

    - Iterative checks on regions of width (w) varying from minimum (sum of required columns of each type) to maximum allowed column span are performed on lists of column indices (Figure 8);
    - Starting from column with the lowest index $i_0 = i_{min}$ the floorplanner checks if enough columns of each type reside between $i_0$ and $i_0 + w$;
    - Incrementations of start index (i) and width (w) take interconnection block distribution into consideration—start and end columns of checked regions are always left- and right-aligned, respectively.

7. Check feasibility of isolation fence insertion and anchoring buffer placement.

    - Depending on selected anchoring points position for partitions, the floorplanner checks if adjacent isolation fence columns (any type) and anchor point CLB columns are available next to selected region (Figure 8);
    - If anchoring buffer placement is impossible, PBlock candidate is removed from the results.

8. Identify column pattern of the region found and group with others.

    - If pattern already exists, regions are appended to associated list;
    - Otherwise, new pattern structure is created.

9. Repeat steps 4–8 for all clock regions.
10. Reduce overlapping patterns.

- Shorter patterns may be entirely contained within longer ones, but they may have more occurrences;
- As overlay floorplanning might be performed before all module variants are designed, minimal size is not always desirable;
- Currently, overlapping pattern reduction supports three strategies: always select minimum size, always select maximum size, and select longer if it does not reduce number of occurrences;
- Intermediate results obtained up to this step can be reused with multiple strategies.

11. Eliminate PBlocks located in areas reserved for routing.

- At least some part of each column must be included in static for the purpose of global routing, as it cannot feed through partitions (due to module relocation restrictions);
- For FPGAs with small number of clock regions (e.g., Artix-7 family, Kintex-7 xc7k160t) default strategy leaves a single central row of clock regions for routing. Custom strategies can be defined;
- This step could be performed at an earlier stage to reduce search space, but the proposed order allows testing multiple strategies of routing interleave placement on single search results (basing on intermediate results from step 10).

12. Eliminate overlapping regions.

- Partition blocks not only cannot overlap but need additional spacing for at least isolation fence and anchoring buffers. Regions that violate this need to be eliminated;
- If a region overlap is identified the procedure keeps the region of the more popular pattern (or instance that occurs earlier in the clock region in the case when both regions use same resource distribution) and eliminates the other—in order to maximize number of PBlocks using the same pattern.

13. Export PBlock candidates, grouped by patterns and ranked by group size.
14. Generate parametric constraints for PE array and reference partitions.

- A parametric, conditional if-clause based constraint file with placement constraints for all partition blocks is generated; only the sections of the file corresponding to PBlocks selected for the overlay are applied to further design.

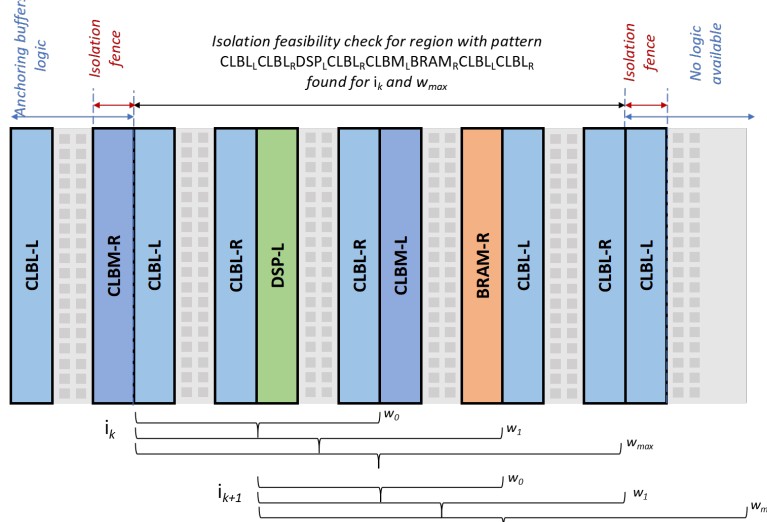

**Figure 8.** Example of iterative region search—required resource columns: BRAM, DSP, 2× logic. If partition ports are located on right or both sides only regions ($i_k$; $i_k + w_1$) and ($i_{k+1}$; $i_{k+1} + w_0$) can be accepted. (Region ($i_k$; $i_k + w_0$) does not satisfy resource requirements—no BRAM. Other regions can only use ports to the left as no logic for anchoring buffers is available to the right).

Pattern ranking determines which overlapping regions are eliminated and what subset of patterns is used in the later stage of the floorplanning. Group size based ranking aims to minimize the number of different PBlock variants used in the design, and consequently the number of required placement and routing runs and partial bitstreams for reprogramming. However, if fully implemented relocatable modules using a particular PBlock construction are available, PBlocks of such construction requires no additional implementation. Therefore, it might be desirable to select them first. For this purpose, support of optional user-defined pattern preferences, that override group based ranking, has been introduced to the proposed floorplanner.

### 3.6. PBlock Candidate Based Floorplanning

The second stage of partition floorplanning consists of solving a combinatorial optimization problem of finding best possible matching. Each of the reconfigurable partitions needs to be individually assigned to one of the PBlock candidates. Considering that module relocation is only possible between compatible regions, it is desirable to use as few PBlock variants as possible, optimally one for all partitions. Starting from the pattern with the most occurrences, if the number of PBlock candidates is greater than or equal to the number of PEs in the overlay, no other PBlock variants are used. Otherwise, PBlock candidates that use pattern with the next largest number of occurrences are added until the condition is fulfilled. PBlocks with different constructions use their individual sets of placement and routing results for all reconfigurable modules in the later implementation stages. Nevertheless, apart from different connection cost resulting from partition ports arrangement, all variants are treated equally for the purpose of PBlock selection.

Finding the optimal matching is virtually impossible due to two factors. First of all, for k reconfigurable partitions and n possible locations (PBlock candidates) there are $n!/(n-k)!$ possible combinations of partition to location assignments. Secondly, a complete evaluation of a solution would require performing a full implementation run with post-routing timing and power analysis, which is extremely time consuming and offers only approximate heuristic-based solution. Moreover, any modifications of static functionality may influence the efficiency of individual partition floorplans. In order to make the floorplanning more feasible, a non-exhaustive search strategy with simplified solution evaluation must be used.

Total wire-length (i.e., a sum of Manhattan distances between directly connected elements) is a parameter commonly used for approximate solution evaluation in the state-of-the art approaches. The total connection length affects average net delays, routing resources utilization, and probability routing congestion. Considering that no routing can feed through relocatable partitions, in the proposed floorplanner a modified distance calculation method that includes the cost of routing detours through other clock regions is used. Additionally, maximal connection length is considered. Being the worst-case scenario, it has significant impact on timing closure unless register-based buffers are used at partition ports. Therefore, the objective function used in the proposed floorplanner comprises of a weighted sum of total wire-length and maximum connection cost with adjustable weights. Since determining the total connection cost requires calculating length of all individual connections, it takes little computational effort to evaluate maximum length too.

The majority of floorplanners (i.e., [40–44]) use simulated annealing search, which is a commonly used strategy for many combinatorial optimization problems. The approach is versatile and offers satisfactory results but requires evaluating an immense number of possible solutions, which makes it slow if objective function is time consuming. Alternatively, a decisive search based on the characteristics of the actual problem can be employed. In theory, similar arrangement of logical and physical representation of partitions can result in a good floorplan. However, the shape of a custom MxN PE array and PBlock candidate distribution rarely match. Nevertheless, some aspects of the geometry can still be exploited to guideline the floorplanning process. The proposed approach is used to determine an initial matching by performing a modified breadth-first search with direction-based penal-

ties and then to attempt to improve the results by performing a local 2-opt optimization between neighboring partitions. The exact procedure is outlined below:

1. Select minimal number of PBlock patterns (as described earlier), identify all PBlock candidates of these patterns, and set them as available for matching.

2. Find the center of PE array and center of mass of PBlock candidate distribution in the FPGA fabric. Start mapping from the center and search outwards.
   - Setting search tree root at the center of PE array reduces the depth of the search;
   - Starting from the center helps to preserve consistency between PE array and the floorplan.

3. Obtain the first set of PEs located closest to the center (same distance in logical network).

4. For each PE, generate hypothetical pairs with all available PBlock candidates.
   - Due to the outward search direction, the number of PEs considered in each iteration grows linearly, but as the number of available PEs decreases at the same rate, the total number of combinations within a single iteration is manageable.

5. Calculate cost of each match based on location (distance, direction) of each candidate.
   - The cost is calculated as the weighted sum of Euclidean distance from the center and Manhattan distance from the directly connected predecessor (if such exists) with added direction-based penalties;
   - Physical location is considered directionally consistent with logical structure if the X and Y components of position in relation to the reference in both PE array and FPGA fabric are not of the opposite sign;
   - A moderate penalty is applied for locations with inconsistent direction in relation to either connected predecessor or the center;
   - A severe penalty is applied if a PE, located in a particular direction from the center, that has s successors (PEs located outwards, considered in later iterations) further in the consistent direction, is paired with a location that has $p < s$ available PBlock candidates that fulfill the direction consistency;
   - The penalties help to preserve consistency between PE array and the floorplan and prevent the otherwise greedy matching from exhausting all candidates in a given direction prematurely. This way, scenarios where outermost elements might be left with no reasonable pairing options can be avoided (Figure 9).

6. Rank pairings for all PEs based on cost. Check for conflicts between best matches. In case of pairing conflicts, check costs of switching to lower ranked options and preserve the matching with the highest cost of change. Repeat until no conflicts occur.
   - At the beginning, all the PEs attempt pairing with their top-rated match;
   - If multiple PEs opt for the same location, only the element that causes the highest growth in cost if moved is preserved, others are re-assigned to lower ranked locations;
   - In the case of movement to an unoccupied location, estimated cost increase is the difference between lower and higher ranked results. If the next best location already has a matching, combined cost of movement to the occupied location and re-assigning the occupant is compared with movement cost to next lower ranked option, the lower value option is used;
   - Re-assignments can lead to new conflicts at different positions. Process is repeated until all conflicts are resolved.

7. Initial floorplanning for this PE group is finished; set selected PBlock candidates as unavailable for next matchings.

8. Obtain the next set of PEs—elements located one position outward in the array—and determine their predecessors (directly connected elements that underwent initial floorplanning). Repeat steps 4–8 for a new set, continue until all PEs are floorplaned.

9. Calculate objective function for the complete initial floorplan. Save individual partial results.

10. Starting from PE [0,0] check if swapping it with any of the neighboring PE (including diagonals) decreases objective function value, if so, accept the swap. Repeat for every node.
    - For a selected PE located at [x,y], feasibility of location swapping with elements at [x+0,y+1], [x+1,y−1], [x+1,y+0], [x+1,y+1] positions is checked. Note that the considered PE is also used for a swap check of previously considered elements;
    - Cost of all NESW connections for new locations of both swapped PEs is calculated and compared to the original. If original or new location is responsible for the maximal connection length, new max is determined.
11. Continue until no PE swaps are performed or until maximum swap limit is reached.
12. For any PE assigned to location with another unused PBlock candidate in the closest proximity, check if swapping with such region decreases objective-function value.
    - Similarly to step 10, NESW connection cost of alternative location is compared to original; swap is only performed if lower cost is obtained.
13. If any changes are performed, go back to step 10, or else save the obtained floorplan.

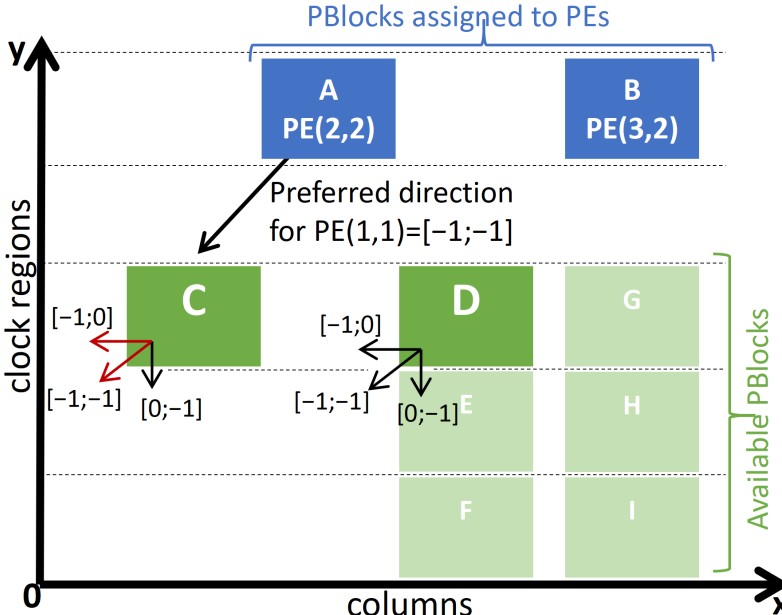

**Figure 9.** Directional consistency in PBlock selection process. In logical array PE(1,1) is located in direction [−1,−1] from its predecessor PE(2,2). To preserve consistent geometry, it should be assigned to a region situated below and to the left of A. However, in the next iteration more elements—PE(0,0), PE(0,1), PE(1,0)—need to be located farther in that direction. Although the position of C in relation to A is consistent with PE(1,1) to PE(2,2) direction, selecting C would result in later placement of elements PE(0,0) and PE(0,1) to the right of PE(1,1)—as no regions are available in their preferred directions. Moreover, if D is used for PE(2,1) or worse PE(3,1), too many elements will opt for PBlocks E and F, resulting in some of them being placed far from their optimal location. Despite immediate directional inconsistency, PE(1,1) should prefer D over C.

## 4. Results

A complex evaluation of the proposed floorplanning solution was performed. Compatibility with Xilinx Vivado tools and general propriety of the operation was tested in static overlay design implementation. Manual floorplans used in the partition relocation tests described in the previous study by the main author of this manuscript [29] were replaced with the results generated by the automated floorplanner proposed in this article. The tool correctly identified identical regions in all tested Xilix 7-Series FPGAs. The automatically generated floorplans were successfully used in the Xilinx Vivado implementation flow.

The quality of results and execution time of the proposed partition to PBlock assignment procedure based on initial modified breadth-first search and fine 2-opt improvements has been evaluated and compared to the simulated annealing approach, which is used in most state-of-the-art solutions. The common objective function was used in both cases. As the results of any stochastic method depend greatly on the randomly selected start point, average values calculated from repeated simulated annealing searches were used to mitigate the influence of this deviation. Additionally, the floorplan found in the modified breadth-first search of the custom solution is used as a starting point for simulated annealing instead of 2-opt in a so-called hybrid approach. For a better quantitative measure of the quality of the generated floorplans, reference objective-function values, calculated as an average for 10 random floorplans for each case, are presented as well.

Tables 1 and 2 present objective-function and elapsed time for the partition to PBlock assignment of various overlays in Artix7 xc7a200t and Virtex7 xc7v2000t, respectively. Results for simulated annealing with constant (30 for A, 100 for B) and PE array size dependent (5xy for C) number of iterations between temperature decrements are presented. Oversizing this parameter results in long execution time, despite procedure termination when a limit of cycles without changes is reached. On the other hand, if too few steps are used in a large search space, the obtained results have worse objective-function values. Further analyses use simulated annealing and scale the parameter with array size, as this approach seems to offer better results. Improvement of the starting point solution used in the hybrid approach has little impact on the execution time and objective function compared to random start point if the same set of parameters is used. The proposed custom search is several orders of magnitude faster than simulated annealing and offers solutions of comparable quality. Large deviations in 2-opt stage execution time are possible, as the procedure is terminated as soon as an iteration with no swaps occurs.

**Table 1.** Comparison of floorplanning methods for Artix7.

| Array Size | | Proposed Approach | | | | Hybrid | | Simulated Annealing | | | | | | Reference Random |
| | | Initial Floorplan | | Optimized Floorplan | | A Steps/dT = 30 | | | | B Steps/dT = 100 | | C Steps/dT = 5xy | | |
| X | Y | $F_{obj}$ | t [ms] | $F_{obj}$ | t [ms] | $F_{obj}$ | t [ms] | $F_{obj}$ | t [ms] | $F_{obj}$ | t [ms] | $F_{obj}$ | t [ms] | $F_{obj}$ |
|---|---|---|---|---|---|---|---|---|---|---|---|---|---|---|
| 2 | 2 | 1860 | 6 | 1860 | 16 | 1594 | 1233 | 1594 | 1274 | 1594 | 8620 | 1657 | 559 | 4933 |
| 2 | 3 | 3695 | 10 | 3693 | 44 | 3322 | 2643 | 3312 | 2741 | 2971 | 26,466 | 3291 | 1740 | 6077 |
| 2 | 4 | 5350 | 11 | 4810 | 41 | 4618 | 6231 | 5453 | 3292 | 4360 | 33,825 | 5617 | 2710 | 10,459 |
| 3 | 3 | 7114 | 15 | 6490 | 46 | 6517 | 7017 | 7332 | 4342 | 6386 | 38,845 | 7561 | 4049 | 11,889 |
| 3 | 4 | 10,599 | 21 | 10,599 | 38 | 9685 | 10,151 | 11,407 | 6263 | 9548 | 45,686 | 11,914 | 6786 | 16,156 |

**Table 2.** Comparison of floorplanning methods for Virtex7.

| Array Size | | Proposed Approach | | | | Hybrid | | Simulated Annealing | | | | | | Reference Random |
| | | Initial Floorplan | | Optimized Floorplan | | A Steps/dT = 30 | | | | B Steps/dT = 100 | | C steps/dT = 5xy | | |
| X | Y | $F_{obj}$ | t [s] | $F_{obj}$ | t [s] | $F_{obj}$ | t [s] | $F_{obj}$ | t [s] | $F_{obj}$ | t [s] | $F_{obj}$ | t [s] | $F_{obj}$ |
|---|---|---|---|---|---|---|---|---|---|---|---|---|---|---|
| 3 | 3 | 10,310 | 0.08 | 10,310 | 0.12 | 5900 | 19.7 | 6,125 | 19.6 | 8106 | 24.4 | 7630 | 13.8 | 27,647 |
| 4 | 4 | 18,856 | 0.21 | 17,211 | 0.44 | 16,945 | 27.5 | 16,112 | 27.3 | 25,042 | 35.2 | 18,423 | 50.5 | 70,060 |
| 5 | 5 | 42,530 | 0.37 | 27,170 | 0.88 | 38,191 | 23.3 | 30,500 | 36.5 | 37,153 | 87.4 | 30,882 | 146.9 | 121,368 |
| 6 | 6 | 60,195 | 0.67 | 43,285 | 1,56 | 57,491 | 31.0 | 53,344 | 48.4 | 53,957 | 149.6 | 45,432 | 305.8 | 160,686 |
| 7 | 7 | 101,480 | 0.93 | 63,231 | 2.57 | 87,675 | 54.8 | 84,454 | 62.3 | 74,346 | 197.8 | 65,674 | 513.8 | 270,153 |
| 8 | 8 | 114,196 | 1.37 | **76,105** | **3.82** | 110,730 | 74.1 | **123,696** | 78.9 | 117,212 | 246.3 | 80,774 | **2263.7** | **337,978** |

The proposed solution seems to offer a significant advantage when floorplanning large arrays. Due to the multitude of potential combinations, simulated annealing based solutions either take significantly more time to finish (for 8 × 8 array: 2263.7 s = 37.7 min compared

to 3.82 s) or introduce the risk of premature termination. In the latter case, minimization of the objective function value is notably less efficient. Although the improvement from the non-optimized reference is still significant (for 8 × 8 array: 337,978 reduced to 123,696), the proposed approach offers much better results ($F_{obj}$ = 76,105).

As the floorplanner attempts to place innermost elements of the array near the center of the fabric, it may not be capable of finding viable solutions located at peripheral positions. Thus, objective-function results for small arrays placed in large devices might be higher than for reference methods. Depending on the connection to I/O ports and various aspects of the design not considered in the objective function, a central location of the array may still be beneficial. Moreover, if placement far from the center is desirable, the central point can be shifted closer to any position by prohibiting regions located in the opposite direction. Considering the significant acceleration offered by the proposed approach, multiple solutions with variously shifted centers can be checked in a fraction of the time required for a single simulated annealing run.

Execution time reduction for designs with small PE arrays might seem irrelevant as floorplanning time of several seconds is negligible in comparison to placement and routing time. However, in the case of larger array sizes, the floorplanning is accelerated by minutes, while offering similar quality results (Figure 10). Considering multiple reiteration during development, this can contribute to a significant time saving. Note that, in order to increase legibility the floorplanning time in Figure 10 is depicted in logarithmic scale. Moreover, as the total cost is geometry dependent ($M(N-1) + N(M-1)$ connections for $M \times N$ array), objective-function values are normalized with respect to total number of connections within PE array.

The proposed floorplanning approach attempts to keep consistency between logical and physical structure. If the geometries of the PE array and partition distribution differ significantly, the method may be less effective. Indeed, differences in the quality of the results can be observed in the floorplanning results for various PE arrays of similar size presented in Table 3.

**Table 3.** Influence of array geometry on the quality of the floorplanning results. Comparison of normalized values of minimization objective function (lower is better).

| X | Y | Nr of PEs | Proposed Approach | Simulated Annealing |
|---|---|---|---|---|
| 3 | 16 | 48 | 1.0863 | 0.7628 |
| 4 | 12 | 48 | 0.9016 | 0.8098 |
| 5 | 10 | 50 | 0.8109 | 0.8992 |
| 6 | 8 | 48 | 0.7320 | 0.8298 |
| 7 | 7 | 49 | 0.7498 | 0.8476 |

The importance of the proposed direction inconsistency penalties is proved by comparing simple outward breadth-first search with the modified version (Table 4). It can be observed that using the proposed modifications allows one to obtain significantly lower objective-function values for both the initial floorplan and 2-opt optimized results.

**Table 4.** Comparison of basic breadth-first search with the proposed modified method, objective-function values (lower is better) for initial and 2-opt optimized floorplans.

| | | Proposed Approach | | Breadth-First Search | |
|---|---|---|---|---|---|
| X | Y | Initial | Optimized | Initial | Optimized |
| 2 | 2 | 3856 | 3856 | 8036 | 5396 |
| 3 | 3 | 10,310 | 10,310 | 22,323 | 11,818 |
| 4 | 4 | 18,856 | 17,211 | 46,267 | 28,352 |
| 5 | 5 | 42,530 | 27,170 | 75,405 | 46,225 |
| 6 | 6 | 60,195 | 43,285 | 113,865 | 77,230 |
| 7 | 7 | 101,480 | 63,231 | 170,516 | 134,301 |

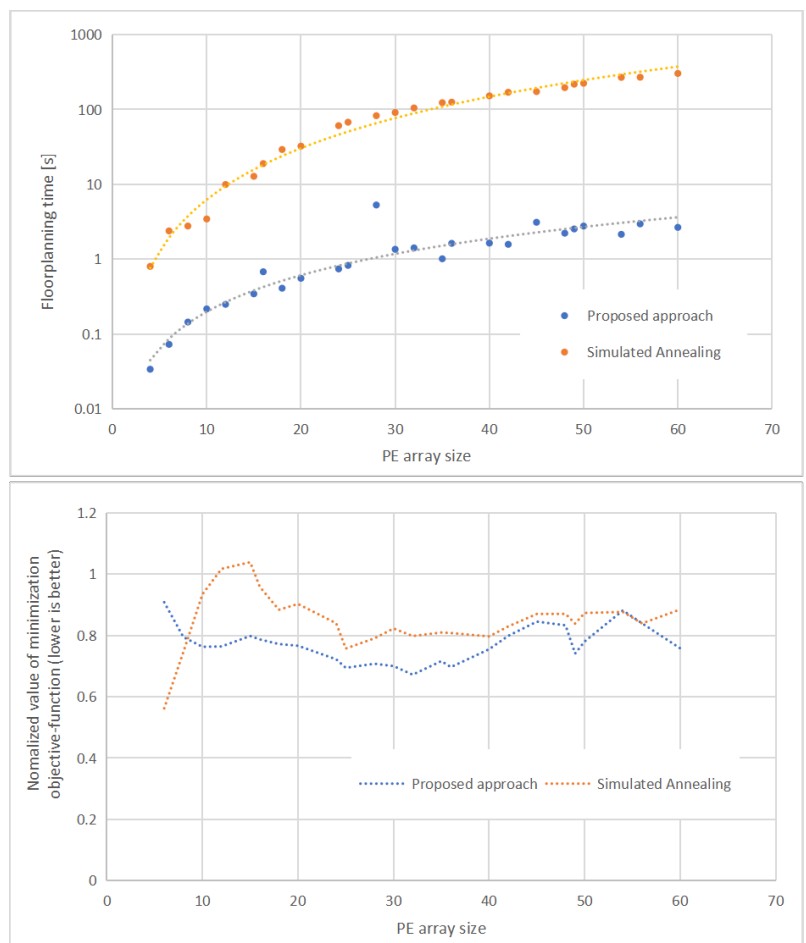

**Figure 10.** Normalized values of minimization objective-function (lower is better) and floorplanning times for different array sizes.

## 5. Conclusions

Virtual Coarse-Grained Reconfigurable Architectures can benefit from partial reconfiguration. If certain components can be reconfigured on their own, reprogramming can be drastically accelerated in comparison to changing the configuration of entire FPGA. Furthermore, an excessive overhead associated with spatial distribution of multiple variants within the same configuration can be avoided. Due to the construction of Xilinx FPGAs, the usage of the mechanism is restricted. The generic VCGRA model proposed in this study presents how and under what assumptions overlay designs can be divided into static and reconfigurable partition parts in order to effectively use partial reconfiguration.

Development productivity can be increased by exploiting module relocation. Same placement and routing results can be reused at multiple positions within the same or across multiple designs. Relocation of partial bitstreams reduces the number of required implementation runs and configuration data storage space. Having numerous instances of identical modules, virtual CGRAs with homogeneous processing elements are prime beneficiaries of these advantages.

Partial reconfiguration implies assigning logical modules to certain physical structures in the FPGA. VCGRAs, based on the most popular 2-D mesh pattern distribution of individual partition blocks in chip, have a crucial impact on routing feasibility and net delays. Taking that into consideration, proper partition floorplanning is essential in such designs, even more so considering that module relocation is only possible between identical partition blocks.

The floorplanning problem for virtual CGRA with relocatable partitions can be divided into two separate tasks—finding homogeneous regions in FPGA fabric and assigning

individual partitions to certain locations. The two-stage solution presented in this study is compatible with Xilinx Vivado tools and capable of automated floorplan generation. The proposed partition to the PBlock assignment approach based on modified breadth-first search with penalties for direction inconsistencies followed by a local 2-opt optimization is drastically faster than state-of-the-art solutions and offers comparable quality of results. Although the difference in the floorplanning time for small arrays might be irrelevant in comparison to placement and routing time, a gain of several minutes can be obtained in the case of larger designs. The offered speedup is crucial considering that VCGRAs are usually used either as prototyping platforms or as a method to increase development productivity in FPGA-based systems.

Although the proposed methodology of mapping CGRAs onto FPGAs with relocatable reconfigurable partitions imposes multiple restrictions on the abstract architecture, it was designed acknowledging trends in CGRA design. Therefore, it should be applicable to many contemporary VCGRAs. In particular, QUKU [4] seems very well fitted to the proposed model, as it uses multiple overlay configurations based on combinations of different variants of processing elements connected in a 2-D mesh.

**Author Contributions:** Conceptualization, Z.M.; methodology, Z.M.; software, Z.M.; data curation, Z.M.; writing—original draft preparation, Z.M.; writing—review and editing, R.K.; supervision, R.K. All authors have read and agreed to the published version of the manuscript.

**Funding:** This research received no external funding.

**Institutional Review Board Statement:** Not applicable.

**Informed Consent Statement:** Not applicable.

**Data Availability Statement:** Not applicable.

**Conflicts of Interest:** The authors declare no conflict of interest.

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
