# Peer review of "Mapping Processing Elements of Custom Virtual CGRAs onto Reconfigurable Partitions"

_electronics, doi:10.3390/electronics11081261_

Round 1

Reviewer 1 Report

General

+ This work build on previous work by the same authors on finding identical regions in an FPGA for relocation of partitions. In this work, they present an improvement over their previous approach, and a new contribution is an algorithm for mapping partitions to specific compatible regions, which shows a considerable improvement over the simulated annealing approach in terms of the time taken to find a solution.

+ This is a very well structured and well written paper, and while I have a few comments, they mostly relate to how some aspects of the work could be highlighted better, and where a cleared distinction could be made between this work and the authors' prior work. Otherwise, I think this is solid work, which I really enjoyed reading and found personally educational.

+ A clearer description of how this work relates to earlier work by the same authors (26 and 36) would be nice. I understand that earlier work was more about (1) finding identical regions, whereas this one focusses on (2) the actual mapping of partitions to certain regions. But there is a degree of overlap, which is fine as it is, but to highlight the contribution of this paper, I think a good idea to really make that unambigious. E.g., how exactly is the approach taken for (1) an improvement over what was done by the same authors earlier? I think just a few lines of comments along these lines would suffice.

+ Related to the previous point, if I understand correctly, the key novelty in this paper is the algorithm expressed in 3.6, and 3.4 is an improvement over a similar approach proposed by the same authors. If such is the case, then 

+ I think it would be useful to articulate the utility of the time saving over the simulated annealing appraoch that this work enables. While we see the numbers there, their significance could be highlighted. E.g., in practice, are some use-cases now in the realm of practical now that the time required for floor-planning so much smaller? The authors do indicate this, e.g. "Rapid floorplanning procedure allows checking multiple solutions 819 with variously shifted center in reasonable time.", but I feel more could be made of this advantage provided by the proposed approach.

# Specific

+ Line 95: Interfacet
+ Line 224: Do you mean to say O(N^2)?
+ Line 341: Two stops. Also inseparability --> inseparable
+ Line 559: ... a the ...
+ Figure 10: Maybe make it cleared that what the y-axis shows, and whether/why lower is better (are we maximizing it or minimizing it?). Same goes for Table 3.

Author Response

Response to Reviewer 1:

Dear Sir/Madam,

Thank you for your insightful review and valuable suggestions.

In the response to justified concerns regarding relations between the presented paper and earlier work by the authors we would like to clarify the issue.

The submitted manuscript presents a continuation of earlier work by the authors ([26,36] in original draft, [29,39] in updated revision). The earlier work was focused on out-of-context implementation and relocation of reconfigurable modules (design constraining, reconfigurable partition buffering, module isolation, applying implementation results as fixed placed and routing constraints, bitstream relocation, etc.). The presented manuscript expands the earlier work in several ways:

- It addresses how the proposed methodology of out-of-context implementation and relocation to can be applied to Virtual CGRAs and presents a feasibility study of such usage (3.2).

- It adds automated floorplanning - both partition compatible region search (3.5) and PBlock selection for individual PEs (3.6). The region search (3.5) is similar to related work by other authors, but introduce several minor improvements (most notably additional isolation fence checking conditions).  The second phase of floorplanning (3.6) is a completely novel approach.

- It reorganizes the modular design flow (3.4) with accordance to the characteristics of VCGRAs and the introduction of automated floorplanning (previously manually generated floorplan file was expected as an additional constraint input).

Explicit clarifications considering the novelty presented in the submitted manuscript have been added to the revised version of the paper (line 521 and line 547).

Following your suggestions, more straightforward interpretation of results presented in the tables, with emphasis on advantages of the proposed approach and their significance in practical use cases have been added in the section “4. Results”.

All the minor issues outlined specifically at the end of your review have been fixed as suggested. Figure 10 have been modified - additional information on Y-axis label was added as requested.

We attach revised version of the manuscript with all changes highlighted in colour.

On behalf of the authors:

Yours faithfully,

Zbigniew Mudza

Reviewer 2 Report

Before anything: Please consider that now partial reconfiguration is dynamic function exchange (DFX). Also please include a table of abbreviations or at the first appearance of the abbreviation please explain. for example: CGRA! The field of reconfiguration or DFX is well known in the field more then twenty years.

In the introduction the the coarse grain and fine grain FPGA-s are described, and the impact of these device on reconfigurable computing. The Dennard scaling and reconfigurable computing should be refereed here. IF one speak about the history of reconfigurable computing then prof. Hartenstein or Patrick Lysaght (Xilinx), etc. In the mean time the technology evolution and the AMD-Xilinx dynamic function exchange should be also mentioned. Also in the introduction Virtual Coarse-Grained Reconfigurable Architectures are explained (here also some references would be welcome). The design steps are also presented.

In the second section the literature review is done. The refereed articles are in detailed analyses and the reconfiguration methods also deeply described with their advantages and disadvantages.

Section 3 presents the available tools and materials with a detailed description and real criticism. Then a proposed architecture is presented with flexible processing elements and highly configurable. In raw 481 is mentioned the the Vivado tool support for reconfiguration. One should mention the from now on the Vitit tools and the adaptive platforms allows VCGRA, true that interconnection between the PEs is not yet detailed described.

In section 3.5 the whole process of design is presented. The authors should mention clearly what is the paper result which can be included in the field of DFX as new results! Then PBlock candidate based floor planning methods is also deeply described. It is not very clear that a procedure or a an automated tool which is the original result. In the section 3.6 the procedure is mentioned, while in the “Results” (777) section “the automated floor planner proposed” is stated. Also please present how the tool can be included in the Xilinx software. Is it available for free? Can be downloaded?

Author Response

Response to Reviewer 2:

Dear Sir/Madam,

Thank you for your insightful review and valuable suggestions. We have introduced the requested improvements to our manuscript (attached to this message) and hopefully improved its quality.

To make sure we address all your concerns regarding our paper, we would like to respond directly to each comment stating what changes have been introduced, clarifying any ambiguity and answering direct questions. All the comments are addressed below (in the Postscriptum). All the changes in the revised manuscript are highlighted in colour in the new PDF document.

We hope that you find our clarifications and improvements in the manuscript satisfactory.

On behalf of the authors:

Yours faithfully,

Zbigniew Mudza

PS.

Point 1: ‘Before anything: Please consider that now partial reconfiguration is dynamic function exchange (DFX).

Response 1: Since Dynamic Function Exchange is a Xilinx specific trade name, we decided to use the term Partial Reconfiguration, as in literature it is used to refer not only to Xilinx-based but also similar solutions by different vendor and even hypothetical FPGAs. Anyway, in order to clarify that partial reconfiguration in Xilinx devices is indeed DFX we added appropriate notice (lines 89 and 379).

Point 2: ‘Also please include a table of abbreviations or at the first appearance of the abbreviation please explain. for example: CGRA! The field of reconfiguration or DFX is well known in the field more then twenty years.’

Response 2: We did not include table of abbreviations as the MDPI template suggests that it is not a common practice for the Electronics journal. We do explain each abbreviation within the text of the article at first occurrence. However, we did not applied this to the abstract, where the unfortunate CGRA abbreviation you refer to was left with no clarification. An explicit explanation is added in the revised version of the abstract.

Point 3: The Dennard scaling and reconfigurable computing should be refereed here. IF one speak about the history of reconfigurable computing then prof. Hartenstein or Patrick Lysaght (Xilinx), etc. In the mean time the technology evolution and the AMD-Xilinx dynamic function exchange should be also mentioned. Also in the introduction Virtual Coarse-Grained Reconfigurable Architectures are explained (here also some references would be welcome).

Response 3: The requested references have been added (lines 21, 27, 44, 67). Partial reconfiguration is now mentioned in the paragraph about evolution of reconfigurable systems in the introduction.

Point 4: ‘In raw 481 is mentioned the the Vivado tool support for reconfiguration. One should mention the from now on the Vitit tools and the adaptive platforms allows VCGRA, true that interconnection between the PEs is not yet detailed described

Response 4: As mentioned in the literature review (“2.2 Virtual Coarse-Grained Reconfigurable Architectures ” and “2.3 Mapping Tools and Techniques”) - although many VCGRA architectures and tools have been presented over the years, they focus on high-level aspects while rarely address the low-level mapping onto a fine-grained physical platform (e.g. efficient usage of gate-level FPGA resources). Since the presented methodology addresses specific, less frequently discussed aspects of VCGRA mapping , we focused on related work and solutions that are the most relevant in comparison with our approach. In particular, the aforementioned line 481 of the original manuscript (“3.3 Module Relocation”) mention low-level relocation of the reconfigurable modules in Xilinx Vivado tools rather than tool support for reconfiguration.

Point 5: ‘The authors should mention clearly what is the paper result which can be included in the field of DFX as new results!’

Response 5: The clarification on the contribution of the paper added in section “3.4. Design Flow”.

Point 6: ‘It is not very clear that a procedure or a an automated tool which is the original result. In the section 3.6 the procedure is mentioned, while in the “Results” (777) section “the automated floor planner proposed” is stated.’

Response 6: In this paper the term “procedure” is used literally as a subroutine/series of steps used to complete a particular task. While the terms “automated floorplanner” and “floorplanning tool” are used solely while referring to our TCL-based implementation of our original methods presented in this paper. We detected and fixed an inconsistency of this naming convention (in chapter “4. Results”), that might have been confusing.

As stated in 3.4 - “The main contribution of this paper is an automated custom floorplanner” based on the original algorithms presented in this paper. (The section has been extended in the revised version to clarify it more explicitly).

The floorplanner is an automated TCL-based tool responsible for two distinctive tasks:

- Finding partition compatible regions.

- Selecting locations for individual PEs.

Exact procedures (subroutines/series of steps) executed by the tool to complete each of these tasks are described in sections 3.5 (partition compatible region search) and 3.6 (partition selection for individual PEs). Both procedures are original. However, while the proposed method of pairing PEs with individual PBlocks (3.6) is completely novel, compatible region search is an improved version of solutions presented in the related work as stated in (2.5).

Point 7:Also please present how the tool can be included in the Xilinx software. Is it available for free? Can be downloaded?’

Response 7: The tool is a set of TCL scripts run using Xilinx Vivado Command Prompt. (Additional explicit statement added - line 550). The tool was created purely for academic purpose and is constantly undergoing improvements and modifications. We have not made it openly available yet. However, we will consider making it public in an open access form.
